# Cassava Fiber Prevents High-Fat Diet-Induced Obesity in Mice Through Gut Microbiota Restructuring

**DOI:** 10.3390/foods14234121

**Published:** 2025-12-01

**Authors:** Yajin Yang, Fuhong Lei, Lily Liu, Yanhong Chen, Qingqing Li, Jieming Long, Zubing Zhang, Aiwei Guo

**Affiliations:** 1College of Biological Science and Food Engineering, Southwest Forestry University, Kunming 650224, China; yangyajin@swfu.edu.cn (Y.Y.);; 2Moringa oleifera Research Center, Yunnan Institute of Tropical Crops, Jinghong 666101, China

**Keywords:** cassava fiber, high-fat diet, obesity, gut microbiota, mice

## Abstract

Cassava fiber (CF) is a novel dietary fiber extracted from cassava by-products. To investigate its anti-obesity mechanism, obesity was induced in mice through a high-fat diet (HFD). Dietary supplementation with 10% CF significantly reduced body weight, body fat, triglycerides, low-density lipoprotein cholesterol, total cholesterol, and fasting blood glucose in mice. CF effectively ameliorated hepatic steatosis and adipocyte hypertrophy, increased the villus height-to-crypt depth ratio, enhanced mucus secretion by intestinal goblet cells, down-regulated the expression of ileal lipid absorption-related genes (*NPC1L1*, *CD36*, and *FABP2*), and up-regulated the short-chain fatty acid receptor GPR43, collectively improving intestinal health. Compared to HFD mice, CF altered the gut microbiota: it increased beneficial Actinobacteria (including *Bifidobacterium* and *Blautia*) and decreased Proteobacteria (including *Desulfovibrio*) (*p* < 0.05). Functional analysis showed that the HFD mice microbiota was enriched in genes linked to disease (e.g., lipid metabolism disorders, cancer, antibiotic resistance), whereas CF-enriched microbiota had genes for energy, carbohydrate, and pyruvate metabolism. Compared to microcrystalline cellulose, CF and MCC both alleviated HFD-induced obesity. In summary, cassava fiber helped prevent obesity in mice by modulating gut microbes, strengthening the gut barrier, and improving host metabolic balance.

## 1. Introduction

Obesity is a global health concern. According to world health organization data, the prevalence of obesity among minors worldwide increased from 2% to 8%, and among adults from 7% to 16% between 1990 and 2022 [1]. Obesity elevates risks of mortality, cardiovascular diseases, metabolic dysfunction, and other comorbidities [2]. Established causative factors include imbalanced diet, physical inactivity, genetic predisposition, medication use, and endocrine disorders [3,4]. The underlying mechanism involves dysregulation of gut endocrine and neurohormonal signaling pathways, leading to increased appetite and energy storage [5]. Dietary modification and exercise remain primary preventive strategies, such as reducing intake of high-fat/high-sugar foods and increasing dietary fiber consumption. Both animal models and human studies have demonstrated the beneficial effects of dietary fiber on obesity and metabolic health [6,7].

Cassava (*Manihot esculenta* Crantz), cultivated broadly across tropical and subtropical zones, has an estimated worldwide production of about 303 million tons [8]. Processing one kilogram of fresh cassava roots into flour and starch yields approximately 0.65 kg of solid residue [9], primarily composed of starch and fiber [10]. The ratio of soluble dietary fiber (SDF) to insoluble dietary fiber (IDF) critically modulates intestinal nutrient absorption and microbiota composition. SDF forms a gel-like matrix in the gut, delaying nutrient transit [11], while being fermented by microbiota to generate short-chain fatty acids (SCFAs), which stimulate enteroendocrine L-cells to secrete glucagon-like peptide-1 (GLP-1) and peptide YY (PYY). GLP-1 enhances insulin secretion, pancreatic β-cell proliferation, and hepatic glycogen synthesis, while promoting satiety [12]. PYY functions to reduce hunger and limit dietary intake [13]. Concurrently, the reticular structure of IDF adsorbs cholesterol, impedes lipid absorption, and augments satiety signals [14]. In our prior study, the extraction process optimization of SC9 (South China No. 9) cassava residue yielded cassava dietary fiber containing 22.92% SDF and 76.14% IDF [15]. This experiment aims to explore the preventive mechanism of the cassava fiber we obtained on high-fat diet-induced obesity in mice, providing a reference for efficient utilization of cassava by-products.

## 2. Materials and Methods

### 2.1. Preparation of Cassava Fiber and Diets Formulation (Diet Composition)

In this study, cassava fiber was extracted from *Manihot esculenta* Crantz cv. CS9 (cultivated in Xishuangbanna, Yunnan Province, China) through a sequential process of washing, enzymatic hydrolysis, and acid treatment. (Extraction conditions: α-amylase concentration 0.8%, temperature 55 °C, pH 6.5, extraction time 3 h). Chemical reagents (Guangdong, Guanghua:) NaOH, sulfuric acid, hydrochloric acid, α-amylase (Novonesis, Denmark). The determination of cassava fiber follows the Determination of Dietary Fiber in Foods (China) [16]. Fiber composition: soluble dietary fiber (SDF) accounts for 22.92%, insoluble dietary fiber (IDF) accounts for 76.14%, and total dietary fiber (TDF) accounts for 99.06%. Four experimental diets were prepared based on Nutrient Requirements of Laboratory Animals [17], as detailed in Table 1.

### 2.2. Animals Experiment

Forty-eight 6-week-old male C57BL/6J mice (22.25 ± 0.98) g were selected. They were randomly divided into four groups, with three cages per group and four mice per cage. The experimental design is illustrated in Figure 1. All mice were provided with food and water ad libitum and housed in a specific pathogen-free (SPF) facility at a constant temperature of 18–22 °C. The trial lasted 16 weeks. Weekly measurements of feed intake and body weight were recorded, and energy intake (kcal) was calculated based on the diet consumption and its energy density (kcal/g). At the conclusion of the study, after a 12 h fast, the mice were euthanized under ether anesthesia via cervical dislocation [18]. Subsequently, plasma, heart, liver, abdominal fat, subcutaneous fat, and intestinal segments were harvested following established protocols. Ileum tissue, cecal contents, and fecal samples were immediately snap-frozen in liquid nitrogen and stored at −80 °C until microbial analysis. This study received ethical approval from the Institutional Animal Care and Use Committee of Southwest Forestry University, with all procedures conducted in accordance with the National Research Council Guide (The Care and Use of Laboratory Animals) [19].

### 2.3. Biochemical and Histological Analysis

Plasma concentration of TG, CHO, HDL, LDL, TP, and GLU were detected by assay kits (Servicebio, GM1113/GM1117) and the automated biochemical analyzers (Leidu Chemray 240, Shenzhen, China). Plasma is clear; hemolysis is excluded. Each sample was measured in triplicate. For histological evaluation, liver, abdominal adipose tissue, and intestinal specimens were fixed in 10% neutral buffered formalin (pH 7.4) for 24 h at 4 °C. After grade alcohol dehydration and xylene clearances, embedding in paraffin and sections in 5 um with a microtome (Leica RM2245, Wetzlar, Germany). The sections of liver and abdomen were used for hematoxylin-eosin (HE) staining and the sections of intestine were used for AB/PAS staining. Microscopic images were captured at 200× and 400× magnifications (Nikon Eclipse E100, Shanghai, China). Image J (Version 1.53t, National Institutes of Health, Bethesda) was used to conduct quantitative analysis of tissue index.

### 2.4. Gene Expression Analysis of Intestinal Lipid-Metabolism Genes

Total RNA was isolated from ileum by TRIzol™ reagent (Servicebio technology Co., Ltd., Wuhan, China) and adjusted to a concentration of 200 ng/μL prior to reverse transcription. Gene expression levels by performing RT-PCR using CFX Connect Realtime PCR Platform (Bio-Rad, Hercules, CA, USA). The primers of intestinal adiponectin are detailed in Table 2. Each targeting gene expression was normalized to GAPDH, and the relative expression of each targeting gene compared to the SD group was calculated using the 2^−△△CT^ method [20].

### 2.5. Gut Microbiota Analysis via 16S rRNA Gene Sequencing

Intestinal contents (Total Genome) DNA was extracted via CTAB method Evaluate DNA concentration and then adjust it to 1 ng/µL with the sterile water: The V3-V4 region of the 16S rRNA gene was amplified by universal primers: 314F (5′-CCTAYGGGRBGCASCAG-3′) and 806R (5′-GGACTACHVGGGTWTCTAAT-3′). The PCR reaction was performed in a mixture with a volume of 25 µL containing: 15 µL Phusion^®^ High-Fidelity PCR Master Mix (NEB). A total of 2 µM of each primer, 10 ng template DNA. Thermocycling denaturing at the initial 98 °C for 2 min, 30 cycles were conducted at 98 °C for 20 s and 50 °C for 30 s and then 72 °C for 30 s and at the end, a 5 min final extension of 72 °C was performed. Amplicons were quantified with the Quant-iT™ PicoGreen™ dsDNA Assay Kit (Invitrogen, Carlsbad, CA, USA) were pooled at equimolar concentrations. Amplicons were purified with the Qiagen Gel Extraction Kit (Qiagen, Hilden, Germany).

Sequence libraries were processed with the MiSeq^®^ DNA PCR-free sample preparation kit (Illumina, San Diego, CA, USA). Library quality was measured using the Qubit^®^ 2.0 fluorescent meter (Thermo Scientific, Waltham, MA, USA) and the Agilent Bioanalyzer 2100 system. In the end, libraries were sequenced on Illumina Novaseq platform with 250bp paired-end reads. Paired-end reads were merged using Flash [21], and raw tags were quality filtered—according to the QIIME v1.9.1 pipeline (Quality threshold is ≤19); tag length filtering: Filter out tags with base lengths shorter than 75% of the tag length. These clean tags were then analyzed again the Silva database through the UCHIME algorithm to identify and eliminate chimeras [22]. Effective tags sharing ≥97% sequence similarity were clustered into operational taxonomic units (OTUs) with Uparse v7.0.1001, and representative sequences from each OTU were taxonomically annotated at the phylum, class, order, family, genus, and species levels using the Silva database. Normalized data were subsequently employed to assess alpha and beta diversity, with principal component analysis (PCA) performed prior to cluster analysis to reduce dimensionality. Alpha diversity indices were calculated using QIIME (version 1.7.0), while principal coordinate analysis (PCoA) was conducted to extract principal coordinates from the multidimensional data. All data analyses was performed through R software (Version 2.15.3).

### 2.6. Statistical Analysis

Animals were randomly assigned to experimental groups and sampled accordingly. The IBM Corp. SPSS 21.0 software was used for analysis. The significance between groups was analyzed using a one-way ANOVA with Turkey’s Multiple Comparison. Data is represented as mean ± sem. The level *p* < 0.05 was considered as significant.

## 3. Results

### 3.1. Cassava Fiber Effectively Inhibited Weight Gain in Mice on a High-Fat Diet

The high-fat diet formulated in this study successfully induced obesity in mice, as demonstrated in Figure 2a. Body weight of mice fed HFD was about 20% greater than SD. From weeks 1 to 16, the growth rate of HFD mice was significantly greater than that of the other three groups, whereas no significant difference was observed between the HFMC and SD groups (Figure 2b). Although food intake varied minimally among groups, energy intake was significantly higher in all three high-fat diet groups compared to the SD group (*p* < 0.01) (Figure 2c,d). The ratios of body weight gain to total body weight in mice fed microcrystalline cellulose and cassava fiber were significantly less than those in the HFD (*p* < 0.01) (Figure 2e). With regard to the assessment of the organs and adipose tissue, there were no significant changes in the heart index or the liver index between the groups (*p* > 0.05) (Figure 2f). However, HFD mice displayed characteristic hepatic pallor suggestive of lipid infiltration, whereas HFMC and HFCF groups maintained normal liver coloration comparable to SD controls (Figure 2a). Mesenteric and subcutaneous fat index in HFD was much higher than the other 3 groups (*p* < 0.01), but there was no difference among HFMC, HFCF, and SD (*p* > 0.05) (Figure 2f).

### 3.2. Cassava Fiber Improves Glycemic Level and Dyslipidemia Caused by High-Fat Diet

Biochemical markers of lipid metabolism are presented in Figure 3. Mice fed HFD had greatly higher levels of lipid TG and LDL than mice fed SD (*p* < 0.01), but the levels of lipid TG and LDL in HFMC and HFCF did not differ from the levels in SD mice (Figure 3a,b). All high-fat groups (HFD, HFMC, HFCF) displayed elevated blood cholesterol (CHO) and glucose (GLU) relative to SD mice (*p* < 0.01), with HFD mice showing maximal increases of 1.73-fold (CHO) and 3.68-fold (GLU) versus SD baseline (Figure 3d,e). Notably, supplementation with microcrystalline cellulose reduced CHO and GLU levels by 23% and 32% compared to the HFD. On the contrary, no considerable differences between the groups regarding HDL and TP can be seen (*p* > 0.05) (Figure 3c,f). So, these results show us that both microcrystalline cellulose and cassava fiber can effectively ameliorate the dyslipidemia and hyperglycemia induced by a high-energy diet in mice.

### 3.3. Cassava Fiber Attenuates Adipocyte Hypertrophy and Enhances Intestinal Mucosal Integrity

Anatomical assessments of liver and white adipose tissues are presented in Figure 4. Liver sections from HFD mice exhibited numerous vacuolar oil droplets—characteristic of fatty liver disease—accompanied by pronounced lymphocyte infiltration indicative of inflammation (Figure 4a). The hepatic lipid droplet area in HFD mice was much bigger than that in other groups (*p* < 0.01) (Figure 4d). Meanwhile, there was no difference among the HFMC, HFCF, and SD. In the mesenteric adipose tissue, the adipocytes from HFD mice had a much higher size than the rest of the groups (*p* < 0.01) (Figure 4b,e), with similar cell sizes observed in the HFMC, HFCF, and SD groups. Additionally, AB/PAS-stained ileal sections revealed that the villus height/crypt depth ratio (V/C) in HFCF mice was way higher than that of HFD mice (*p* < 0.01; Figure 4c,f–h). Concurrently, the secretion of both neutral and acidic mucus by the villi and basal epithelial cells was markedly reduced in HFD mice compared to the other groups (*p* < 0.01; Figure 4c,i), while similar mucus production was noted between the HFMC and HFCF groups. These findings suggest cassava fiber effectively suppresses HFD-driven adipocyte hypertrophy while promoting mucin-mediated intestinal mucosal barrier enhancement.

### 3.4. Cassava Fiber Affects the Expression of Genes Related to Intestinal Lipid Metabolism

The relative expression levels of ileal epithelial cells with lipid metabolism-related gene are presented in Figure 5. In contrast to SD mouse ileum, FFAR2 (GPR43) expression was notably downregulated by HFD (*p* < 0.05); and FABP2, CPC, and CD36 expression was significantly upregulated by HFD (*p* < 0.05). However, in mice supplemented with microcrystalline cellulose or cassava fiber, the expression levels of these genes did not differ from those in the SD group. In addition, there was no significant difference in the MGAT2 expression of all groups. And such kinds of discoveries indicate that cassava fiber probably does help regulate lipid metabolism through mitigating high-energy diet-induced alterations in intestinal gene expression.

### 3.5. Cassava Fiber Modulates Gut Microbial Composition

Cecal microbiota profiling via V3-V4 16S rRNA gene sequencing (Illumina NovaSeq platform) revealed distinct microbial community structures across dietary groups (97% similarity threshold for OTU clustering). Richness and evenness were calculated by alpha diversity analysis; at the OTU level the dispersion of alpha-diversity within the HFD mice was lower than other groups. The HFCF mice showed a similar amount as well as dispersion of species when compared to SD (Figure 6a). Chao 1 and shannon indices also showed the HFCF were similar to the SD in richness and evenness of diversity (Figure 6b,c). Furthermore, principal component analysis (PCA) revealed that the microbial community structure in HFD mice differed and segregated from SD (Figure 6d), while cluster analysis indicated that the cecal microbiota of HFCF mice closely resembled that of SD mice. The results indicated that high-fat diet-induced obesity, as well as the intervention with cassava fiber, significantly alters gut microbiome composition, which may be pivotal in regulating and preventing obesity.

Different dietary group mice had different abundance of cecum microbiota. Despite differences in diet composition, their gut bacteria’s main taxa were Actinobacteria, Firmicutes, Verrucomicrobia, Proteobacteria, and Bacteroidetes. Phylum level (Figure 6e,f), the relative abundance of Actinobacteria was much less in HFD mice compared to HFCF mice (*p* < 0.05) (Figure 6j), whereas the relative abundance of Proteobacteria in HFD mice was much more than in HFMC and HFCF groups (*p* < 0.01) (Figure 6k). Correspondingly, at the genus level (Figure 6g,h), SCFA-specific *Bifidobacterium* (Actinobacteria) were significantly more abundant at the genus level in HFCF mice than in HFD mice (*p* < 0.01) (Figure 6 m) and HFMC mice, but pro-inflammatory *Desulfovibrio* (Pro-bacteria) were significantly more abundant in HFD mice than in HFMC mice (*p* < 0.05) (Figure 6l). Also, the relative amount of *Blautia* in HFD and HFCF mice was dramatically less when compared with HFCF mice (*p* < 0.001) (Figure 6n). Even if many of the microbial taxa did not show statistical significance in the phylum and genus abundance, the above differences in the flora with specific metabolic functions suggest that cassava fiber mediates anti-obesity effects through microbial metabolite reprogramming.

According to 16S rRNA gene-sequence information of the samples microbiota we annotated, gut microbiota functions by mapping homologous sequences to the KEGG database. Tax4Fun analysis revealed that, compared to SD and HFCF mice, the gut microbiota of HFD mice enriched KEGG homologs associated with human diseases at Level 1 (Figure 7a,d). Correspondingly, at level 2, we found genes involved with lipid metabolism, cancer, and drug resistance were enriched (Figure 7b,e). In contrast, at level 3, the gut microbiota of HFCF mice was significantly overrepresented in functional genes involved in energy metabolism, carbohydrate metabolism, and pyruvate metabolism (Figure 7c,f). It means different dietary plans might impact gut health differently since they alter the specific gene sets performing functions in the gut microbiome.

## 4. Discussion

Obesity is widely recognized as a global public health issue facing the human population. We usually refer to the imbalance between the caloric intake and energy utilization as the cause of obesity [23]. Obesity is closely associated with more than 50 diseases, including diabetes, high cholesterol, gallbladder disease, and metabolic dysfunction, and increases the risk of death [2]. In the diets of countries such as Eastern Europe, China, and France, lard is often used in cooking to enhance the flavor of food. These animal fats are high in saturated fatty acids (SFAs), and eating too much of these kinds of fats for long periods of time is related to weight gain and heart problems [24]. Our research proved that feeding 35% lard to adult mice led to obesity, and this obesity was shown through an increase in the mice’s weight, an increase in belly and subcutaneous fat, and through pathological manifestations of cardiovascular disease and fatty liver, with total triglyceride, LDL (low density lipoprotein), cholesterol, and glucose levels all at elevated levels, with many fat vacuoles and lymphocytes present in liver cells. The direct reason is that dietary saturated fatty acids are transported into intestinal epithelial cells for further processing and packaging to form chylomicrons, which are transferred to metabolic tissues through the lymphatic and circulatory systems [25], resulting in obesity.

Nutritional approaches to preventing obesity include adjusting dietary patterns and increasing dietary fiber intake. There are many studies which have shown that dietary fiber can prevent obesity by regulating intestinal digestion and absorption, microflora composition, enzyme activity, short-chain fatty acids, and other related receptor (GPCRs, Y2), promoting hormone secretion (PYY, GLP-1, NF-κB) and regulating appetite, metabolism, and immunity through the brain-intestinal axis [26,27,28]. And, through our study, we found that MCC and cassava fiber also help with obesity and reduce the TG, LDL, CHO, and GLU in the blood of mice ingesting a high-fat diet. MCC is a purified polysaccharide derived from natural plant cellulose polymerized with β-1, 4-glucoside bond. It is used as food raw material and dietary fiber in the food industry. MCC has been proven to absorb lipids to reduce blood lipids, affect the expression of enzymes in lipid metabolism, and have a positive effect on gastrointestinal physiology [29]. The cassava fiber in this study was extracted from South China No. 9 (SC9), a high yield and low cyanoside variety widely cultivated in Yunnan Province, China. The proportions of soluble and insoluble fibers in cassava fiber were 22.92% and 76.14%, respectively. The gel-like substance formed by soluble fiber in the gastrointestinal tract can affect digestion and absorption of the contents [30]. Insoluble fiber could increase fecal bulk and decrease transit time [31]. The study by Lin et al. also pointed out that people with higher fiber intake and those with a higher proportion of increased soluble fiber had lower BMIs [32]. A few reviews and an analyses of the studies that have been conducted have identified that fiber intake can give a measure of protection from cardiovascular illness [33] and type 2 diabetes [34]. First, the fiber-rich cell wall acts as a barrier stopping food from being digested and so uses less energy and CHO [35]. Second, soluble fiber increases the viscosity of chyme, which impedes the α-activity of amylase and the diffusion of glucose through the epithelium cell [36]. Indeed, from the indices of view of anti-obesity indicators, the effect of cassava fiber is also better than MCC.

In addition to fiber being able to dilute energy and shorten transport time, the resistance to obesity it provides is also reflected in the structure and function of the intestine [37]. Changes in the height of intestinal villi and the depth of intestinal recess can directly affect the surface area of intestinal contact with food and determine the absorption and digestion of nutrients [38]. It is pointed out in this study that the MCC and cassava fiber can promote an increase in villus height and V/C value in the ileum through the dietary fiber fermentation process, the production of SCFAs in the intestine, the supply of energy to the intestinal epidermis cell, and the promotion of the proliferation of crypt cells and goblet cells to increase the villi height [39]. Furthermore, MCC and cassava fiber can promote the differentiation of goblet cells and effectively alleviate the obstruction of intestinal epithelial mucus secretion caused by the high-fat diet. Other studies [40] also suggest that the mucus layer is structurally dependent on Mucus—2 recombinant protein (MUC2) synthesized by goblet cells, and absence of fiber results in a thinned mucus layer and damages our intestinal health.

Lipid-associated transporters in intestinal epithelial cells are key to fat absorption. The Niemann-Pick C1-like 1 gene (NPC1L1) controls the expression of a membrane protein on the brush membrane of the small intestine epithelium [41], a target for intestinal cholesterol absorption. NPC1L1 transports cholesterol to the endoplasmic reticulum in the intestinal cells, which is esterified into cholesterol ester (CE) by cholesterol esterification enzyme (ACAT2), and then forms chylomicrons that are transported to the blood circulation [42]. Polymorphism of the NPC1L1 gene are related to plasma total cholesterol and low-density lipoprotein cholesterol (LDL-C) levels. FABP2 (intestinal FABP) is one of the intracellular protein families highly expressed in intestinal villi epithelial cells, and it binds strongly with long-chain fatty acids (LCFA) [43]. FABP2 is a major regulatory of lipid metabolism within cells and tissue. FABP2 transports fatty acids into intestinal epithelial cells to synthesize triglycerides [44], and its over expression will increase the transport of fatty acids and cause diseases such as dyslipidemia and obesity [45]. Gajda et al. [46] confirmed that FABP2 knockout mice lost weight after a high-fat diet, considering that FABP2 is associated with lipid uptake in the small intestine. In addition, FABP2 collaborates with the fat transporter (CD36) to promote the transmembrane absorption of long-chain fatty acids [47] and the production of chylomicron [48]; free fatty acids cause post translation variations in these transporters [49]. And highly expressing all three of these kinds of transmembrane protein genes can enable intestinal epithelial cells to absorb and transform a large number of lipids. Similarly to our results, the gene expressions of *NPC1L1*, *FABP2*, and *CD36* transmembrane proteins in the ileal epithelial cells of HFD mice are significantly higher than those of the control mice, and dietary fiber can effectively reduce the expression of these lipid transport genes. The results were also confirmed by TG, LDL, and CHO in the blood of the mice.

In contrast, HFD reduced *FFAR2* (GPR43) gene expression in mouse ileal epithelial cell membranes. GPR43, GLP-1, and PYY are all co expressed and have similar roles by way of regulation of appetite and insulin action in L Intestional cells to control energy homeostasis [50,51,52]. Among them, the PYY is released by the intestinal epithelial cells in response to the presence of lipids in the intestinal lumen and it controls the production of FABP2 by the L-cells [53]. GPR43, PYY, and GLP-1 on intestinal epithelial cell membrane can be activated by SCFAs [54,55], and FFARs have been reported to have physiological functions such as facilitation of insulin and incretin hormone secretion [56]. Ge et al. [57] showed that intraperitoneal injection of sodium acetate in mice could activate GPR43 and rapidly reduce the level of plasma FA in vivo. Similarly, in our study, dietary fiber under a high-fat diet may also regulate these lipid transporters through intestinal microbial fermentation to produce SCFAs, which regulates the absorption of fat and energy by intestinal epithelial cells. This positive regulatory effect is also reflected in the blood indices of mice.

The changes in the composition of gut microbiota caused by diet are closely associated with metabolic disorders and obesity [58,59]. The core microbiota in the animal intestine mainly consists of Firmicutes, Actinobacteria, Verrucomicrobia, Proteobacteria, and bacteroidetes [60]. In our study, the significant enrichment of *Blautia* within the phylum Firmicutes in the intestines of HFCF mice, compared to HFD and HFMC mice deprived of insoluble fiber, drew our attention. Among mammals, *Blautia* is a potential probiotic with obesity-regulating effects, following *Akkermansia* [61]. In obesity research in mice and humans, *Blautia* shows a negative correlation with several biomarkers related to obesity and metabolic disorders in multiple studies [62,63,64]. In addition, eating orally *Blautia wexlerae* has been shown to prevent obesity and diabetes induced by a high-fat diet in mice [65]. Adding corn fiber to a high-fat diet can increase the abundance of *Blautia* in mouse feces and improve obesity [66]. Enrichment of *Blautia* was also observed in the feces of rats when fiber was added to a high-cholesterol diet [67]. In our study, the inhibitory effects of cassava fiber on obesity and hyperlipidemia may also be associated with the enrichment of *Blautia*. In contrast, microcrystalline cellulose had minimal effects. Parmar et al. [68] found that *Blautia* isolated from the rumen showed prominent metabolic capabilities for insoluble fibers. This may suggest that the fermentation substrates of *Blautia* are specific. The regulation of intestinal microorganisms to host is mainly mediated by metabolites. The main metabolic product of *Blautia* in the intestine is acetate [69], which, by activating the GPR41 and GPR43, inhibits insulin signaling and fat accumulation in adipocytes, thereby promoting lipid and glucose metabolism and alleviating obesity [70]. Similarly to our results, the high expression of the GPR43 gene in the intestinal epithelium of HFD-fiber mice may also be contributed by the high abundance of *Blautia*. Some in vitro experiments have also confirmed that *Blautia* can prevent obesity by esterifying hydroxylated long-chain fatty acids [71] and metabolizing tryptophan to produce indole-3-acetic acid (I3AA) [62]. I3AA restrained liver lipogenesis via negatively controlling the expression of the FAS and SREBP-1c, which encoded fatty acid synthetases and SREBP-1c, respectively [72]. These studies on *Blautia* will provide valuable information for future microbiome-based strategies for the early prevention of obesity and its complications.

In addition, the differences in the gut microbiota of mice induced by cassava fiber intervention were also reflected in the following: fiber deprivation (HFD) led to a decrease in the abundance of *Bifidobacterium* within the phylum Actinobacteria and an increase in the abundance of *Desulfovibrio* within the phylum Proteobacteria. Similarly, in the gut Microbiota of 66 obese and diabetic patients, a low abundance of *Bifidobacteria* (Actinobacteria) and a high abundance of Proteobacteria were observed [60]. Fiber diet intervention can increase the abundance of *Bifidobacterium adolescentis* and protect male mice from diet-caused obesity [73]. *Bifidobacterium pseudocatenulatum*’s newly found evidence, its strains have an endo-1, 4-β-xylanase which is a gene to metabolize plant-based long-chain xylan (a form of insoluble diet fiber) [74], producing short-chain fatty acids (lactate, acetate) associated with host health [75]. At the same time, Tax4Fun showed that the gut microbiota of cassava fiber mice enriched genetic information related to energy metabolism (Level 2) and pyruvate metabolism (Level 3), which corresponded to the enrichment of *Bifidobacterium*. Because *Bifidobacterium* contains an important metabolic gene-pyruvate-kinase in its 16S rRNA [76], it can convert pyruvate into lactic acid under anaerobic conditions [77]. In addition, *Bifidobacterium pseudocatenulatum* promotes the biosynthesis of secondary bile acids by producing bile salt hydrolase [78], whereas *Bifidobacterium pseudolongum* produces more of acetate which in turn can block IL-6/JAK1/STAT3 pathway using GPR43 [79], both of which reduce excessive fat deposition in the body. These diet-induced differences in gut microbiota, despite the specificity between strains. But overall, intervention with dietary fiber and its associated floras could be a way to tackle metabolic conditions such as obesity and diabetes.

Tax4Fun functional prediction can quickly predict the potential metabolic pathways and functions of a microbial community by mapping to the KEGG functional database, without the need for complete genome data. It is worth noting that, compared to cassava fiber mice, Tax4Fun showed that HFD mice enriched genetic information related to human diseases (Figure 7a,d), lipid metabolism, and drug resistance (Figure 7b,e), which corresponded to the enrichment of the *Desulfovibrio*. There is more energy which leads to there being more Desulfovibrio, which damages the host. Lots of studies previously identified that *Desulfovibrio* in gut microbiota contributes to obesity and diabetes [80,81]. The *Desulfovibrio*-derived H_2_S, richly found in the gut microbiota of metabolic syndrome, can inhibit mitochondrial respiration and suppress the secretion and gene expression of GLP-1 in the intestinal L-cells of mice [82]. This metabolic disruption through the gut-brain axis leads to obesity and impaired glucose tolerance [83]. Additionally, *Desulfovibrio* can desulfate mucin, thereby facilitating Prevotella in degrading intestinal mucin and disrupting the mucus barrier [84]. In our study, HFD mice with enriched desulfation showed reduced mucin secretion in the intestinal epithelial AB/PAS staining. These results are associated with the abundance of *Desulfovibrio*. Of course, such results reflect trends rather than actual gene expression levels. We need to validate these findings in subsequent studies by integrating metagenomic or transcriptomic data.

## 5. Conclusions

Our study successfully established a high-fat mouse model with a formulated high-fat diet, which exhibited marked obesity and related lesions. Under a high-fat diet, both cassava fiber and MCC effectively prevent obesity, maintain homeostasis in fat, liver, and intestinal epithelial cells, and regulate serum glucose and lipid levels. Furthermore, cassava fiber may alter *Bifidobacterium*, *Blautia*, and *Desulfovibrio* abundance in the intestine, affecting gut barrier integrity and host metabolic processes, resulting in an anti-obesogenic effect. Of course, in future research it is necessary to carry out metagenomic and metabolomic detection of SCFAs to further verify the mechanism of preventing obesity.

## Figures and Tables

**Figure 1 foods-14-04121-f001:**
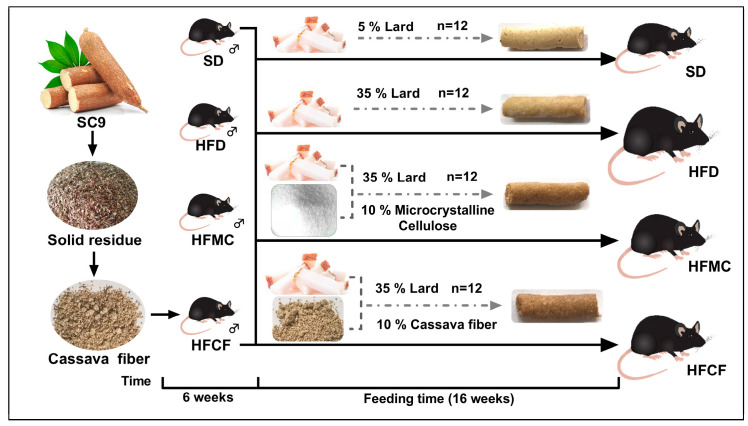
Schematic diagram of experiment.

**Figure 2 foods-14-04121-f002:**
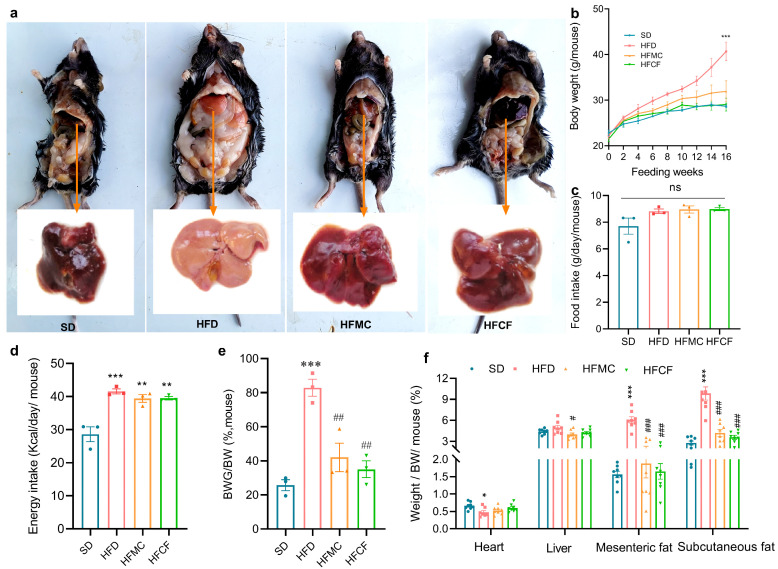
Abdominal anatomical features and liver morphology, body weight, and dietary indicators of 22-week-old mice under different diets (SD: standard diet; HFD: high-fat diet; HFMC: high-fat diet with 10% microcrystalline cellulose; HFCF: high-fat diet of 10% cassava fiber; n = 12/group); (**a**) abdominal anatomical features and liver morphology; (**b**) body weight trajectory, measurements taken per mouse; (**c**) food intake, measured by cage; (**d**) energy intake (kcal/mouse/day), measured by cage; (**e**) body weight gain ratio (final weight—initial weight)/final weight), measured by cage; (**f**) organo index (organ weight / body weight × 100%), six mice were randomly selected from each cage as samples. Data represent mean ± SEM. * Statistical annotation: * *p* < 0.05, ** *p* < 0.01, *** *p* < 0.001 compared with the SD; # *p* < 0.05, ## *p* < 0.01, ### *p* < 0.001 compared with the HFD.

**Figure 3 foods-14-04121-f003:**
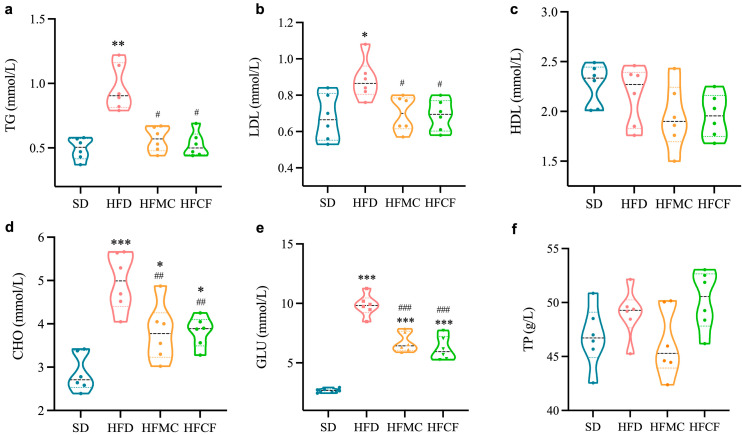
Levels of biochemical indices of lipid levers in the serum of mice. n = 6/group. (**a**) Triglycerides (TG); (**b**) low-density lipoprotein cholesterol (LDL); (**c**) high-density lipoprotein cholesterol (HDL); (**d**) cholesterol (CHO); (**e**) glucose (GLU); (**f**) total protein (TP). Data represent mean ± SEM. * Statistical annotation: * *p* < 0.05, ** *p* < 0.01, *** *p* < 0.001 compared with the SD; # *p* < 0.05, ## *p* < 0.01, ### *p* < 0.001 compared with the HFD.

**Figure 4 foods-14-04121-f004:**
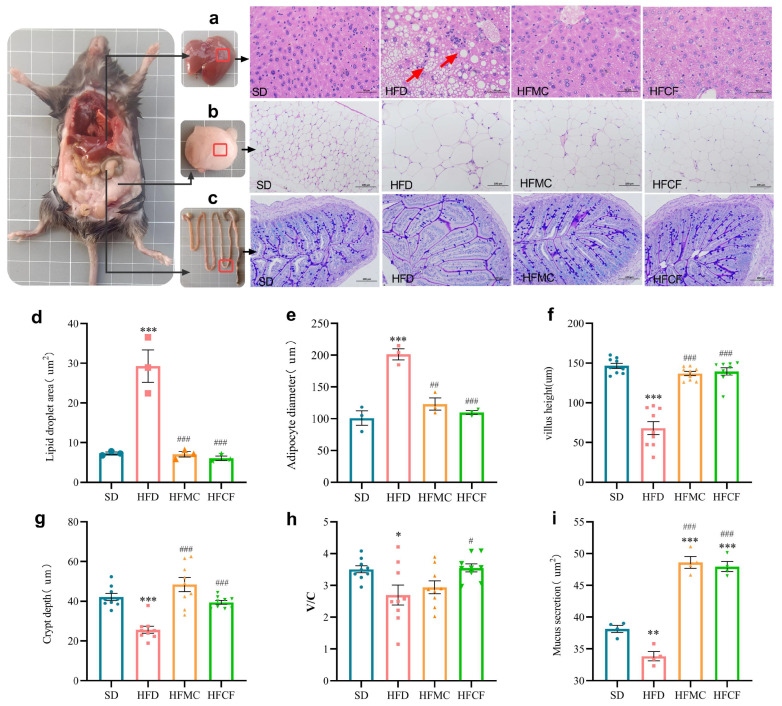
Histopathological characteristics of liver and adipose tissue and their quantitative analysis. (**a**) H&E-stained liver sections (400×); (**b**) H&E-stained mesenteric adipose tissue (200×); (**c**) AB-PAS-stained ileal sections (200×); (**d**) hepatic lipid droplet area; (**e**) mesenteric adipocyte diameter; (**f**) ileal villus height; (**g**) ileal crypt depth; (**h**) villus height/crypt depth (V/C) ratio; (**i**) neutral and acidic mucus of ileal tissue. Histological measurements were performed on 3 sections per mouse group, with >100 adipocytes counted per field of view. Data represent mean ± SEM. * Statistical annotation: * *p* < 0.05, ** *p* < 0.01, *** *p* < 0.001 compared with the SD; # *p* < 0.05, ## *p* < 0.01, ### *p* < 0.001 compared with the HFD.

**Figure 5 foods-14-04121-f005:**
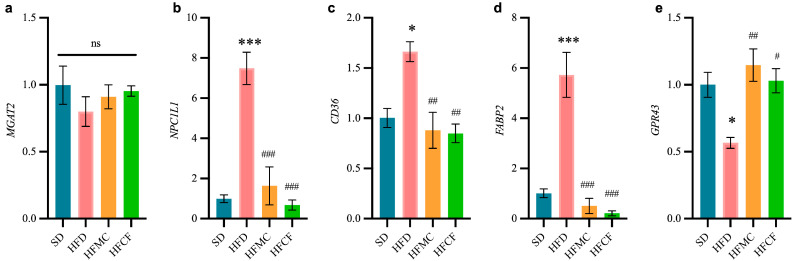
Relative transcript levels of ileal genes involved in lipid metabolism in ileum (2^−ΔΔCt^). n = 6/group. (**a**) Monoacylglycerol acyltransferase 2 (MGAT2). (**b**) Niemann-pick C1-like 1 (NPC1L1). (**c**) Fatty acid translocase (CD36/FAT). (**d**) Intestinal fatty acid-binding protein 2 (FABP2). (**e**) Free fatty acid receptor 2 (FFAR2/GPR43). Data represent mean ± SEM. * Statistical annotation: * *p* < 0.05, *** *p* < 0.001 compared with the SD; # *p* < 0.05, ## *p* < 0.01, ### *p* < 0.001 compared with the HFD.

**Figure 6 foods-14-04121-f006:**
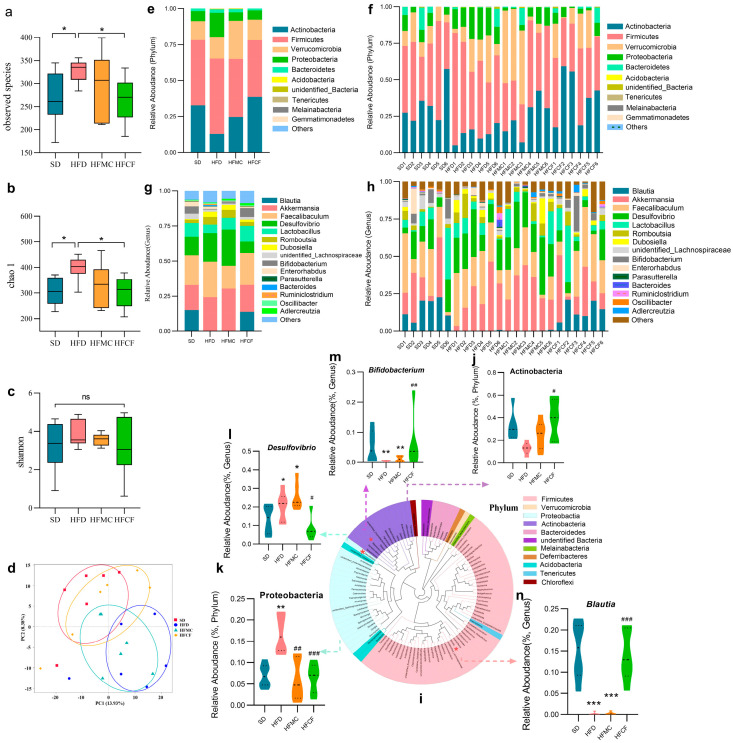
Alpha and beta diversity of the microbiota in the cecal contents and classification analysis of the microbiota. n = 6/group. Data represent mean ± SEM. * Statistical annotation: * *p* < 0.05. (**a**) Observed species between groups. (**b**) Alpha diversity chao1 index. (**c**) Alpha diversity shannoon index. (**d**) Principal component analysis between groups. (**e**,**f**) Phylum-level relative abundance. (**g**,**h**) Genus level relative abundance. (**i**) Phylogenetic tree of predominant genera. (**j**,**k**) Differential flora at phylum level. (**l**–**n**) Differential flora genus levels. Data represent mean ± SEM. * Statistical annotation: * *p* < 0.05, ** *p* < 0.01, *** *p* < 0.001 compared with the SD; # *p* < 0.05, ## *p* < 0.01, ### *p* < 0.001 compared with the HFD.

**Figure 7 foods-14-04121-f007:**
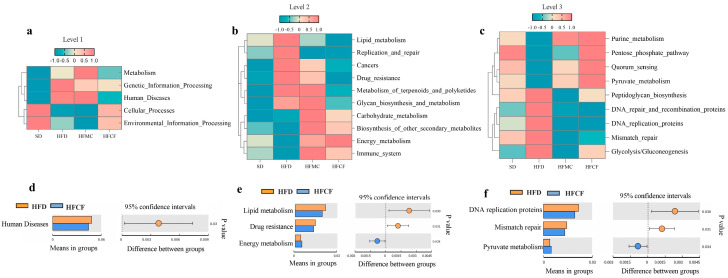
Hierarchical functional profiling of gut microbiota via Tax4Fun prediction. n = 6/group. (**a**–**c**) KEGG ortholog distribution across functional hierarchies. (**d**–**f**) Differentially enriched pathways (a/d: level 1. b/e: level 2. c/f: level 3).

**Table 1 foods-14-04121-t001:** Ingredient composition and calculated nutrient levels of the mice diets (%, air-dried basis).

Items 1	SD	HFD	HFMC	HFCF
**Ingredients (%)**				
**Corn Starch**	55.06	18.64	14.18	14.01
**Fish meal (65% CP)**	13.84	14.01	13.83	14.00
**Lard**	5.00	35.00	35.00	35.00
**Casein**	10.00	10.00	8.89	8.75
**Saccharose**	10.00	10.00	10.00	10.00
**Wheat Meal**	5.00	5.00	5.00	5.13
**B** **entonite**	0.00	5.00	0.00	0.00
**Microcrystalline Cellulose**	0.00	0.00	10.00	0.00
**Cassava Fiber**	0.00	0.00	0.00	10.00
**L-Lysine**	0.00	0.09	0.69	0.70
**DL-Metonine**	0.30	0.35	0.50	0.50
**Limestone**	0.10	0.01	0.01	0.01
**Dicalcium Phosphate**	0.10	0.30	0.30	0.30
**Sodium Chloride**	0.20	0.20	0.20	0.20
**Cholesterol**	0.00	1.00	1.00	1.00
**Choline Chloride (50%)**	0.10	0.10	0.10	0.10
**Vitamin and Mineral Premix 2**	0.30	0.30	0.30	0.30
**Total**	100.00	100.00	100.00	100.00
**Calculated Nutrient Content 3**				
**ME (kcal/kg)**	3714.06	4708.30	4708.30	4713.08
**Crude Protein (%)**	18.32	18.32	18.33	18.32
**SDF (%)**	0.04	0.04	0.04	2.33
**IDF (%)**	0.31	0.31	10.31	7.93
**TDF (%)**	0.34	0.34	10.34	10.26
**Lysine (%)**	1.55	1.56	1.52	1.54
**Methionine (%)**	0.88	0.87	0.84	0.84
**Calcium (%)**	0.63	0.64	0.63	0.64
**Nonphytate Phosphorus (%)**	0.51	0.51	0.50	0.50

1. SD: standard diet; HFD: high-fat diet; HFMC: supplemented with 10% microcrystalline cellulose; HFCF: supplemented with 10% cassava fiber. 2. The premix supplied the following per kilogram of diet: Vitamin A, 8000 IU; Vitamin D_3_, 1000 IU; Vitamin E, 20 mg; Vitamin K_3_, 0.5 mg; Vitamin B_1_, 2 mg; Vitamin B_2_, 8 mg; Vitamin B6, 3.5 mg; niacin, 35 mg; biotin, 0.18 mg; Folic acid, 0.55 mg; Pantothenic acid, 10 mg; Manganese, 100 mg; Zinc, 80 mg; Iron, 80 mg; Copper, 9 mg; Iodine, 0.35 mg; Selenium, 0.15 mg. 3. Calculated according to the actual analyses of the raw materials.

**Table 2 foods-14-04121-t002:** Primer sequence for Realtime qPCR.

Gene	Forward Primer	Reverse Primer
MGAT2	ACTGAAGCAGCAGGAGTGTC	GTCAAGGCTAGCCCCATGTT
NPC1L1	TGAGGACCTTTGCCTTGACC	GTTTCGGTGGGGGCAGATT
CD36	GGAACTGTGGGCTCATTGCT	CAACTTCCCTTTTGATTGTCTTCTC
GPR43	TGTTCAGTTCCCTCAATGCCA	CAGGATTGCGGATCAGTAGCA
FABP2	GAGCTCGGTGTAAACTTTCCCT	CCTCTCGGACAGCAATCAGC
GAPDH	CCTCGTCCCGTAGACAAAATG	TGAGGTCAATGAAGGGGTCGT

## Data Availability

The original contributions presented in this study are included in the article. Further inquiries can be directed to the corresponding authors.

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
