# Peer review of "Cassava Fiber Prevents High-Fat Diet-Induced Obesity in Mice Through Gut Microbiota Restructuring"

_foods, 2025, doi:10.3390/foods14234121_

Round 1
Reviewer 1 Report
Comments and Suggestions for Authors
-
Page 2, Line 41: The plant name should be italicized to follow scientific naming conventions.
-
Page 2, Line 43: A reference is needed to support the statement.
-
Page 2, Line 54: The abbreviation "SC9" should be defined upon its first appearance for clarity.
-
Page 2, Section 2.1: The plant name should be italicized throughout this section.
-
Page 2, Section 2.1: The enzymatic extraction process should be described in more detail, and an appropriate reference should be provided. Additionally, the first mention of "CS9" should be elaborated, even though it is discussed later in the manuscript.
-
Page 2, Section 2.1, Lines 62–65: The categorization of fibers as "detergent fiber" is not appropriate in this context. It should be revised to "soluble dietary fiber," "insoluble dietary fiber," etc., in line with current nutritional terminology.
-
Page 2, Section 2.1, Lines 62–65: The chemical characterization methodology should be elaborated, and relevant references should be cited to support the approach.
-
Page 3, Section 2.2, Lines 76–77: The sentence is difficult to follow due to excessive information being condensed. The phrase “six-weeks-age” should be corrected to “six-week-old.” The description of experimental groups and replicates is unclear and requires simplification for better readability.
-
Page 3, Section 2.2, Lines 82–85: A reference should be cited to support the methodology described.
-
Page 4, Line 105: For "ImageJ software," please include the version number and supplier/source information.
-
Formatting Issue: Throughout the manuscript, there is a lack of spacing between words and in-text citations. This issue also applies to figure references. While some instances are highlighted, the manuscript should be carefully reviewed and corrected throughout.
-
Figure 6i: A higher-resolution image is required, as the current version makes the text difficult to read.
-
Page 10, Line 264: The abbreviation "SCFA" should be defined upon first use.
-
Figure 7 (a–f): These images should be replaced with higher-resolution versions to improve legibility.
-
Page 11, Line 329: A reference is needed to support the statement.
-
Page 12, Line 339: There is a spelling error in the highlighted text that should be corrected.
-
Page 14, Line 440: Another spelling error is present in the highlighted text and should be corrected.
-
References Section: The reference list requires revision. Several formatting issues have been highlighted, but a thorough review is recommended.
-
Major Scientific Concern: While the manuscript discusses the potential of cassava fiber in preventing obesity in mice, it lacks any analysis of short-chain fatty acids (SCFAs) in body fluids, which are likely to be produced from cassava fiber fermentation. Furthermore, the discussion section does not address the types of SCFAs previously reported in similar studies, nor does it correlate the findings with existing literature on dietary fiber composition, SCFA production, and their role in obesity. This is a significant gap that should be addressed to strengthen the scientific validity and relevance of the study.

Author Response
Dear reviewers and editor;
On behalf of my co-authors, we thank you very much for giving us an opportunity to revise our manuscript, we appreciate editor and reviewers very much for their positive and constructive comments and suggestions on our manuscript entitled “Cassava fiber prevents high-fat diet-induced obesity in mice through gut microbiota restructuring”(ID: foods-3994255). Please see our response below to the reviewers' comments and also yellow highlighted parts in manuscript for correction. The reference order has also been updated in the new manuscript according to the revisions.
Comments and suggestions for the author, along with details of the author's revisions:
Reviewer #1 :
- Page 2, Line 41: The plant name should be italicized to follow scientific naming conventions.
Author: Thank you very much for your constructive comments on our manuscript. The plant names in the article have been corrected and are now displayed as italics. Revised manuscript (section 1).
- Page 2, Line 43: A reference is needed to support the statement.
Author: Thanks for your suggestion. We have added references to support the statement. Revised manuscript (section 1).
- Page 2, Line 54: The abbreviation "SC9" should be defined upon its first appearance for clarity.
Author: We defined it when “SC9 ”first appeared. Revised manuscript.
- Page 2, Section 2.1: The plant name should be italicized throughout this section.
Author: Thanks for your suggestion. The plant names in the article have been corrected and are now displayed as italics. Revised manuscript (section 2.1).
- Page 2, Section 2.1: The enzymatic extraction process should be described in more detail, and an appropriate reference should be provided. Additionally, the first mention of "CS9" should be elaborated, even though it is discussed later in the manuscript.
Author: Extraction conditions have been added to the text. Revised manuscript (section 2.2)
- Page 2, Section 2.1, Lines 62–65: The categorization of fibers as "detergent fiber" is not appropriate in this context. It should be revised to "soluble dietary fiber," "insoluble dietary fiber," etc., in line with current nutritional terminology.
Author: Thank you very much for your constructive comments on our manuscript. The original terms “detergent fiber” and “insoluble detergent fiber” have been revised to “soluble dietary fiber” and “insoluble dietary fiber”. Revised manuscript (section 2.1).
- Page 2, Section 2.1, Lines 62–65: The chemical characterization methodology should be elaborated, and relevant references should be cited to support the approach.
Author: Thanks for your suggestion. We have added the extraction method for cassava fiber and relevant references in section 2.1.
- Page 3, Section 2.2, Lines 76–77: The sentence is difficult to follow due to excessive information being condensed. The phrase “six-weeks-age” should be corrected to “six-week-old.” The description of experimental groups and replicates is unclear and requires simplification for better readability.
Author: Thanks for your suggestion. This sentence has been rewritten as: Forty-eight 6-week-age male C57BL/6J mice (22.25 g±0.98 g) were selected. They were randomly divided into four groups, with three cages per group and four mice per cage. Revised manuscript (section 2.2)
- Page 3, Section 2.2, Lines 82–85: A reference should be cited to support the methodology described.
Author: Thanks for your suggestion. We have added references to support the anesthetic handling methods in mice. Revised manuscript (section 2.2).
- Page 4, Line 105: For "ImageJ software," please include the version number and supplier/source information.
Author: Thanks for your suggestion. We have added the version number and source information for the Image J software. Revised manuscript (section 2.3).
- Formatting Issue: Throughout the manuscript, there is a lack of spacing between words and in-text citations. This issue also applies to figure references. While some instances are highlighted, the manuscript should be carefully reviewed and corrected throughout.
Author: Thanks for your suggestion. We have revised the lack of spacing between words and text citations throughout the entire document.
- Figure 6i: A higher-resolution image is required, as the current version makes the text difficult to read.
Author: Thanks for your suggestion. We have increased the font size in part of Image 6 and enhanced its resolution. Revised manuscript, figure 6, has been uploaded to the system with the original image at a resolution exceeding 300 dpi.
- Page 10, Line 264: The abbreviation "SCFA" should be defined upon first use.
Author: Thanks for your suggestion. We have added a definition where the abbreviation “SCFA” appears. Revised manuscript (section 3.5)
- Figure 7 (a–f): These images should be replaced with higher-resolution versions to improve legibility.
Author: Thanks for your suggestion. We have increased the font size in part of figure 7 (a–f) and enhanced its resolution. Revised manuscript, figure 7, has been uploaded to the system with the original image at a resolution exceeding 300 dpi.
- Page 11, Line 329: A reference is needed to support the statement.
Author: Thanks for your suggestion. We have added citations at this location to support the statement.
- Page 12, Line 339: There is a spelling error in the highlighted text that should be corrected.
Author: Thank you very much for your advice. The incorrect spelling of “studie” has been corrected to “studies”. Revised manuscript.
- Page 14, Line 440: Another spelling error is present in the highlighted text and should be corrected.
Author: This was our oversight; the incorrect spelling of “mic” has been corrected to “mice”. Revised manuscript.
- References Section: The reference list requires revision. Several formatting issues have been highlighted, but a thorough review is recommended.
Author: Thanks for your suggestion. We have reviewed and revised the other references based on the formatting issues you raised.
- Major Scientific Concern: While the manuscript discusses the potential of cassava fiber in preventing obesity in mice, it lacks any analysis of short-chain fatty acids (SCFAs) in body fluids, which are likely to be produced from cassava fiber fermentation. Furthermore, the discussion section does not address the types of SCFAs previously reported in similar studies, nor does it correlate the findings with existing literature on dietary fiber composition, SCFA production, and their role in obesity. This is a significant gap that should be addressed to strengthen the scientific validity and relevance of the study.
Author: Thanks for your suggestion. Our study did not include the measurement of metabolites in intestinal contents, which is a limitation. We plan to address this issue in future research.
Additionally, the manuscript's discussion section incorporates literature on the types of short-chain fatty acids (SCFAs) and their roles in relation to dietary fiber composition and obesity, as reported in previous similar studies. For example: The discussion highlights SCFAs' crucial regulatory role on lipid absorption-related genes in intestinal epithelial cells (Paragraph 4 of discussion). It also explores the foundational research demonstrating how Blautia and Bifidobacteria utilize dietary fiber to produce SCFAs that regulate lipid absorption genes.

Reviewer 2 Report
Comments and Suggestions for Authors
Dear Authors,
The manuscript with title "Cassava fiber prevents high-fat diet-induced obesity in mice through gut microbiota restructuring" is interesting and deals with very important nutritional problem - obesity. However, this manuscript should be improved in several points:
- If authors studied the impact of cassava fiber on the obesity in mice, why they do not include examination of insulin in their study?
- Impact of the cassava fiber on insulin resistance should be very important in their final conclusions.
- Line 60-62: how extraction of cassava fibers was performed, which solvents was used?
- The first part of Materials and methods should be reagents and protocols.
- Which is the importance of bentonite in mice feed?
- Which group of the dominant microbial taxa in the mice gut microbiota was the most affected by the diet?
Author Response
Dear reviewers and editor;
On behalf of my co-authors, we thank you very much for giving us an opportunity to revise our manuscript, we appreciate editor and reviewers very much for their positive and constructive comments and suggestions on our manuscript entitled “Cassava fiber prevents high-fat diet-induced obesity in mice through gut microbiota restructuring”(ID: foods-3994255). Please see our response below to the reviewers' comments and also yellow highlighted parts in manuscript for correction. The reference order has also been updated in the new manuscript according to the revisions.
Comments and suggestions for the author, along with details of the author's revisions:
Reviewer #2 :
The manuscript with title "Cassava fiber prevents high-fat diet-induced obesity in mice through gut microbiota restructuring" is interesting and deals with very important nutritional problem - obesity. However, this manuscript should be improved in several points:
- If authors studied the impact of cassava fiber on the obesity in mice, why they do not include examination of insulin in their study?
Author: Thank you for suggesting insulin level testing, which is crucial for mouse obesity studies. Our initial research design primarily focused on evaluating cassava fiber's effects on energy intake, weight gain, fat accumulation, and gut microbiota. Although we did not directly measure insulin, our findings observed significantly improved fasting blood glucose levels with cassava fiber (Figure 4e), suggesting enhanced glucose homeostasis. Furthermore, the observed reductions in adipose tissue weight and hepatic steatosis are common indirect indicators of improved insulin sensitivity.
Based on these findings, we delved deeper into the gut microbiota structure-liver lipid metabolism pathway as a downstream mechanism, constituting the primary innovation of this study. Consequently, research priorities and limited experimental resources led us to focus primarily on these aspects.
We acknowledge that the lack of direct serum insulin level measurements and insulin tolerance testing represents a limitation of this study, restricting our ability to comprehensively elucidate cassava fiber's action pathways. Following your suggestion, we have listed “effects on insulin signaling pathways” as a key indicator for future research.
- Impact of the cassava fiber on insulin resistance should be very important in their final conclusions.
Author: Thanks for your suggestion. We have incorporated the relevant conclusions regarding cassava fiber's role in preventing obesity and regulating blood glucose and lipid metabolism into our conclusions section.
- Line 60-62: how extraction of cassava fibers was performed, which solvents was used?
Author: We have added a brief description of the cassava fiber extraction process in the Materials and Methods section: washing, enzymatic hydrolysis, and acid treatment (Extraction conditions: α-amylase concentration 0.8%, temperature 55℃, pH 6.5, duration 3 h). Detailed fiber extraction procedures are described in our separate, forthcoming article.
- The first part of Materials and methods should be reagents and protocols.
Author: Thanks for your suggestion. We have added the relevant reagents for fiber extraction to the Materials and Methods section. The experimental protocol includes the diet formulation and the grouping design.
- Which is the importance of bentonite in mice feed?
Author: To design different fat and fiber additions across groups while maintaining a constant total ingredient ratio (100%), we referenced the National Standard for Laboratory Animal Feeds. Adjustments were made based on the nutritional levels of each ingredient to ensure that only ME (kcal/kg) and fiber (%) differed in the Calculated nutrient content (Table 1, note 3). Bentonite is a non-nutritive feed additive that functions physically (as a binder) without introducing additional variables.
- Which group of the dominant microbial taxa in the mice gut microbiota was the most affected by the diet?
Author: In our findings, distinct dietary compositions yielded different microbial abundances. A high-fat diet increased the relative abundance of Proteobacteria (Desulfovibrio) in the mouse gut, while dietary fiber elevated the abundance of Actinobacteria (Bifidobacteria) and Blautia. Related literature indicates that metabolites produced by these microbial populations are closely associated with gut homeostasis and the absorption and conversion of nutrients.

Reviewer 3 Report
Comments and Suggestions for Authors
Major scientific concerns (summary)
Diet formulation & confounding. Table 1 shows the HFD includes 5% bentonite whereas HFMC/HFCF replace this with 10% fiber; casein also differs across HF groups. These changes mean groups differ by more than “fiber type,” complicating causal attribution. Energy density is reported as essentially equal across all HF diets despite adding 10% low‑calorie fiber—this needs justification.
Fiber composition inconsistencies. Reported ADF (76.93%) exceeds NDF (72.43%), which is not possible because ADF is a subset of NDF. Also, “soluble detergent fiber (SDF)” is likely a mislabel of soluble dietary fiber; please harmonize fiber terminology and provide AOAC‑TDF values.
Experimental unit & statistics. Animals were housed in “three replicates with four mice per replicate” per group, but analyses appear to treat individual mice as independent for cage‑level measurements (e.g., feed/energy intake). Body‑weight trajectories need repeated‑measures/mixed models. Multiple comparisons across taxa/pathways require FDR control.
Microbiome methods inconsistencies. The text alternates between V4 and V3–V4 regions and between MiSeq/HiSeq/NovaSeq platforms; read length reporting is inconsistent. Tax4Fun inferences are over‑interpreted as “genes linked to disease,” which needs softer language and statistical control.
Mechanistic overreach. Claims about SCFAs/GLP‑1/PYY and “gut barrier strengthening” are not directly measured; statements should be tempered or supported by added data (e.g., SCFA quantification, GLP‑1/PYY assays, MUC2 expression).
Terminology & copy errors. GPR43 is a receptor, not a transporter; “adipocyte hyperplasia” is used where the data show hypertrophy; several genus‑abundance statements mis‑name the comparator group; “ABS staining” appears instead of AB/PAS.
Line 17, please, change “short-chain fatty acid transporter GPR43” to “short‑chain fatty acid receptor GPR43 (FFAR2).”
Line 23, I recommend, either provide direct HFMC‑vs‑HFCF statistics across all primary outcomes to substantiate “CF more effectively alleviated obesity than MCC,” or soften the claim to “CF and MCC both alleviated…”.
Line 29–57, please consider adding a concise gap analysis: what is unknown about cassava fiber specifically (composition, fermentability, distinct effects vs cellulose), why 10%, and what mechanistic hypothesis is being tested beyond general “fiber is beneficial.”
Line 54–55, please, clarify whether 22.92% (SDF) + 76.14% (IDF) reflects dietary fiber fractions; reconcile totals and ensure methods (AOAC vs detergent system) are stated consistently.
Line 60–66, please, provide full extraction parameters for cassava fiber (enzyme types and units, pH, temperature, time, acid identity and concentration, solvent ratios) so the material can be reproduced.
Line 62–64, I recommend, correct “soluble detergent fiber (SDF)” to “soluble dietary fiber (SDF)” and reconcile fiber nomenclature across SDF/IDF vs NDF/ADF systems.
Line 63–64, must be corrected: ADF (76.93%) cannot exceed NDF (72.43%); please re‑analyze and correct Table 1 values.
Table 1 (Lines 66–74), please, justify why bentonite (5%) is present only in HFD but not in HFMC/HFCF; consider including a matched inert filler across all high‑fat diets to isolate the fiber effect.
Table 1 (Lines 66–74), it would be beneficial to explain how ME (MJ/kg) remains identical despite 10% fiber inclusion; specify the energy calculation method and whether diets were formulated isoenergetically.
Table 1 (Lines 66–74), please consider reporting total dietary fiber (TDF, AOAC) for all diets; “crude fiber” underestimates fiber and makes HFMC (10.20%) vs HFCF (5.17%) difficult to interpret.
Line 75–83, please, define the experimental unit (cage vs mouse) for each endpoint, indicate whether randomization and blinding were applied, and specify whether feed intake was measured per cage or per mouse.
Line 81–86, I recommend, reconsider “ether anesthesia” wording in light of current animal‑care guidance; at minimum, justify and confirm IACUC approval for this approach.
Line 93–105, please consider adding kit catalog numbers, sample handling details (fasting verification, hemolysis exclusion), and numbers of technical replicates for each biochemical assay.
Section 2.4 heading (Line 106), please, change “Gene expression analysis of adiponectin (Adipoq)” to “Gene expression analysis of intestinal lipid‑metabolism genes,” since Adipoq was not measured.
Lines 110–114, I recommend, provide full primer sequences (without hyphenation breaks), amplicon sizes, annealing temperatures, and primer efficiencies; confirm GAPDH stability across diets.
Lines 115–131, must be improved: resolve V4 vs V3–V4 inconsistency and MiSeq/HiSeq/NovaSeq platform discrepancies; state one platform, read length, and vendor/facility.
Lines 131–142, please, detail sequence‑quality thresholds (Phred cutoffs), chimera reference set, rarefaction depth/normalization, number of samples per group post‑QC, and include negative controls. Consider updating to QIIME 2.
Line 143–148, I recommend, (i) declare normality/homoscedasticity checks; (ii) use mixed models for repeated body‑weight measures; (iii) clarify whether Tukey was applied to all pairwise contrasts; and (iv) use FDR for multi‑taxon/pathway tests.
Lines 150–166 & Figure 2 (p. 6), please, report n per group analyzed, the experimental unit for energy intake (likely cage), and provide pairwise statistics within HF groups; define “body weight gain ratio” (the current formula is unclear).
Lines 176–187 & Figure 3 (p. 7), please, specify whether the reported 23% and 32% reductions in CHO/GLU refer to HFMC or HFCF; provide effect sizes and exact p‑values for HFMC vs HFCF.
Lines 196–212 & Figure 4 (p. 8), please, replace “adipocyte hyperplasia” with “hypertrophy,” since only diameter was measured; add the number of cells/fields quantified per mouse and whether analysis was blinded.
Lines 205–211 & Figure 4 (p. 8), I recommend, quantify goblet cell density and/or MUC2 expression to substantiate “enhanced mucus secretion,” or soften the claim to “increased AB/PAS staining.”
Lines 221–229 & Figure 5 (pp. 8–9), please, confirm that MGAT2, NPC1L1, CD36, FABP2, and GPR43 were measured in ileal epithelium; consider adding protein‑level validation for key targets.
Lines 236–256 & Figure 6 (pp. 9–10), please consider replacing “dispersion of species diversity was lower” with specific alpha‑diversity metrics (e.g., Shannon, Chao1) and report statistical tests used.
Lines 260–272 & Figure 6 (pp. 9–10), must be corrected: the text states “Bifidobacterium in both HFD and HFCF was lower than HFCF,” which is self‑contradictory; this should read “HFD and HFMC were lower than HFCF.” Similarly fix the Blautia sentence.
Lines 273–286 & Figure 7 (p. 10), I recommend, temper Tax4Fun interpretations (“genes linked to disease”) and apply multiple‑testing correction; present effect sizes for pathway differences and avoid equating predicted functions with measured gene content.
Lines 309–337 (Discussion), please consider avoiding causal statements about SCFAs/GLP‑1/PYY unless measured; propose this as a hypothesis or add SCFA and hormone data.
Line 338, please, correct the typo “Other studie[36]” to “Other studies [36].”
Lines 382–401 (Discussion), I recommend, distinguish clearly between correlational abundance changes (e.g., Blautia) and causal effects; when citing external intervention studies, indicate strain/species and dosage to avoid over‑generalization.
Lines 409–427 (Discussion), please consider aligning claims about Bifidobacterium with the actual genus/species detected and again avoid causal language unless supported by targeted assays.
Line 431–444, please, change “intestinal epithelial ABS staining” to “AB/PAS staining.”
Conclusions (Lines 446–451), I recommend, tempering to “CF attenuated HFD‑associated phenotypes…” and add a sentence acknowledging diet formulation limitations and the need for SCFA/hormone measurements.
Figures (pp. 6–10), please consider adding sample sizes (n) on each panel, clarifying statistical tests used for the symbols (*, #), and ensuring axis labels/units are explicit (e.g., “kcal/mouse/day”).
Data availability, I recommend, depositing raw 16S reads in a public repository (e.g., SRA) and providing the accession number in the manuscript to ensure reproducibility.
Author Response
Dear reviewers and editor;
On behalf of my co-authors, we thank you very much for giving us an opportunity to revise our manuscript, we appreciate editor and reviewers very much for their positive and constructive comments and suggestions on our manuscript entitled “Cassava fiber prevents high-fat diet-induced obesity in mice through gut microbiota restructuring”(ID: foods-3994255). Please see our response below to the reviewers' comments and also yellow highlighted parts in manuscript for correction. The reference order has also been updated in the new manuscript according to the revisions.
Comments and suggestions for the author, along with details of the author's revisions:
Reviewer #3 :
- Diet formulation & confounding. Table 1 shows the HFD includes 5% bentonite whereas HFMC/HFCF replace this with 10% fiber; casein also differs across HF groups. These changes mean groups differ by more than “fiber type,” complicating causal attribution. Energy density is reported as essentially equal across all HF diets despite adding 10% lowcalorie fiberthis needs justification.
Author: To design different fat and fiber additions across groups while maintaining a constant total ingredient ratio (100%), we referenced the National Standard for Laboratory Animal Feeds. Adjustments were made based on the nutritional levels of each ingredient to ensure that only ME (kcal/kg) and fiber (%) differed in the Calculated nutrient content (Table 1, note 3). Bentonite is a non-nutritive feed additive that functions physically (as a binder) without introducing additional variables. Thus, despite differing ingredient ratios, the levels of other nutrients were balanced as much as possible across all groups.
- Fiber composition inconsistencies. Reported ADF (76.93%) exceeds NDF (72.43%), which is not possible because ADF is a subset of NDF. Also, “soluble detergent fiber (SDF)” is likely a mislabel of soluble dietary fiber; please harmonize fiber terminology and provide AOACTDF values.
Author: Thank you very much for your constructive comments on our manuscript. In mouse and human nutrition studies, SDF and IDF components are typically used in dietary fiber, while ADF and NDF are primarily applied in ruminant nutrition research. Therefore, we have removed the ADF and NDF data from the our manuscript. In addition,the terms “soluble detergent fiber” (SDF) and “insoluble detergent fiber” in the manuscript have been corrected to “soluble dietary fiber” (SDF) and “insoluble dietary fiber”.
- Experimental unit & statistics. Animals were housed in “three replicates with four mice per replicate” per group, but analyses appear to treat individual mice as independent for cagelevel measurements (e.g., feed/energy intake). Bodyweight trajectories need repeatedmeasures/mixed models. Multiple comparisons across taxa/pathways require FDR control.
Author: Thanks for your suggestion. Each replicate in our experimental design corresponds to one cage, and we have updated the term “replicates” to “cages” in section 2.2. The experimental data in Figures 3c-d were measured per cage and converted to per-mouse data based on the number of mice per cage. Figures 3b and 3e were measured in units of individual mice and analyzed based on the average value of the cages(replicates).
- Microbiome methods inconsistencies. The text alternates between V4 and V3–V4 regions and between MiSeq/HiSeq/NovaSeq platforms; read length reporting is inconsistent. Tax4Fun inferences are overinterpreted as “genes linked to disease,” which needs softer language and statistical control.
Author: Thanks for your suggestion. Inconsistencies in sequencing platforms were our oversight. These have been verified and corrected in sections 2.5 and 3.5. Library generation: MiSeq® DNA PCR-Free Sample Preparation Kit was used. Library sequencing: Illumina NovaSeq platform.
As you mentioned, Tax4Fun maps taxonomic information onto known microbial genomes through phylogenetic relationships, thereby inferring the potential functional capabilities of microbial communities. We have added the following description to our end of discussion: “Of course, such results reflect trends rather than actual gene expression levels. We need to further validate these findings in subsequent studies by integrating metagenomic or transcriptomic data.”
- Mechanistic overreach. Claims about SCFAs/GLP1/PYY and “gut barrier strengthening” are not directly measured; statements should be tempered or supported by added data (e.g., SCFA quantification, GLP1/PYY assays, MUC2 expression).
Author: Thanks for your suggestion. Our results did not directly address the conclusion that SCFAs/GLP-1/PYY and other bioactive compounds strengthen the gut barrier. The discussion section referenced prior studies indicating these bioactive compounds are associated with gut microbiota, which served to indirectly explore their relevance to our experimental findings.
- Terminology & copy errors. GPR43 is a receptor, not a transporter; “adipocyte hyperplasia” is used where the data show hypertrophy; several genusabundance statements misname the comparator group; “ABS staining” appears instead of AB/PAS.
Author: Thanks for your suggestion. Yes, GPR43 is a receptor. We have corrected”transporter” to “receptor”, amended “adipocyte hyperplasia” to “adipocyte hypertrophy” (Section 3.3), and rectified the erroneous “ABS staining” to “AB/PAS staining” (Section 4).
- Line 17, please, change “short-chain fatty acid transporter GPR43” to “short chain fatty acid receptor GPR43 (FFAR2).”
Author: Thanks for your suggestion. Same as question 6: GPR43 is a receptor. We have corrected “transporter” to “receptor” .
- Line 23, I recommend, either provide direct HFMC vs HFCF statistics across all primary outcomes to substantiate “CF more effectively alleviated obesity than MCC,” or soften the claim to “CF and MCC both alleviated…”.
Author: Your suggestion is well-founded. We have revised the statement “CF more effectively alleviated obesity than MCC” to read “CF and MCC both alleviated obesity...”
- Line 29–57, please consider adding a concise gap analysis: what is unknown about cassava fiber specifically (composition, fermentability, distinct effects vs cellulose), why 10%, and what mechanistic hypothesis is being tested beyond general “fiber is beneficial.”
Author: Thanks for your suggestion. Yes, HFMC and HFCF both exert positive preventive effects against the obesity phenotype in mice. We state in our conclusion: “Under a high-fat diet, cassava fiber and MCC both effectively prevent obesity, maintain homeostasis in fat...”
- Line 54–55, please, clarify whether 22.92% (SDF) + 76.14% (IDF) reflects dietary fiber fractions; reconcile totals and ensure methods (AOAC vs detergent system) are stated consistently.
Author: Thanks for your suggestion. In accordance with AOAC standards, we have more accurately redefined the original terms “soluble detergent fiber (SDF)” and “insoluble detergent fiber (IDF)” as “soluble dietary fiber (SDF)” and “insoluble dietary fiber (IDF)”. These constitute the primary components of dietary fiber. (Section 2.1)
- Line 60–66, please, provide full extraction parameters for cassava fiber (enzyme types and units, pH, temperature, time, acid identity and concentration, solvent ratios) so the material can be reproduced.
Author: Thanks for your suggestion. We added reagents and parameters for cassava fiber extraction in Section 2.2.
- Line 62–64, I recommend, correct “soluble detergent fiber (SDF)” to “soluble dietary fiber (SDF)” and reconcile fiber nomenclature across SDF/IDF vs NDF/ADF systems.
Author: Thanks for your suggestion. Same as question 10, we have more accurately redefined the original terms “soluble detergent fiber (SDF)” and “insoluble detergent fiber (IDF)” as “soluble dietary fiber (SDF)” and “insoluble dietary fiber (IDF)”. (Section 2.1)
- Line 63–64, must be corrected: ADF (76.93%) cannot exceed NDF (72.43%); please reanalyze and correct Table 1 values.
Author: Thanks for your suggestion. We have verified that the ADF and NDF data are incorrect, while the SDF and IDF values are correct. In mice nutrition studies, SDF and IDF components are typically used, while ADF and NDF are primarily applied in ruminant nutrition research. Therefore, we have removed the ADF and NDF data from the manuscript.
- Table 1 (Lines 66–74), please, justify why bentonite (5%) is present only in HFD but not in HFMC/HFCF; consider including a matched inert filler across all high fat diets to isolate the fiber effect.
Author: Thanks for your suggestion. To design different fat and fiber additions across groups while maintaining a constant total ingredient ratio (100%), we referenced the National Standard for Laboratory Animal Feeds. Adjustments were made based on the nutritional levels of each ingredient to ensure that only ME (kacl/kg) and fiber (%) differed in the Calculated nutrient content (Table 1, note 3). Bentonite is a non-nutritive feed additive that functions physically (as a binder) without introducing additional variables. Thus, despite differing ingredient ratios, the levels of other nutrients were balanced as much as possible across all groups.
- Table 1 (Lines 66–74), it would be beneficial to explain how ME (MJ/kg) remains identical despite 10% fiber inclusion; specify the energy calculation method and whether diets were formulated is isoenergetically.
Author: Thanks for your suggestion. Metabolizable energy (ME) in feed formulations is typically an estimated value calculated based on raw material nutrient content (Nutrient Requirements of Laboratory Animals. 4th Revised ed). It generally reflects the energy level of the feed and is primarily used for formulation purposes, it may still vary from the actual ME of the formulated feed due to factors such as raw material sourcing and processing. Our subsequent research will consider measuring actual nutrient levels. Additionally, in Table 1, we converted the units and values for ME to kcal to align with the energy intake data for mice.
- Table 1 (Lines 66–74), please consider reporting total dietary fiber (TDF, AOAC) for all diets; “crude fiber” underestimates fiber and makes HFMC (10.20%) vs HFCF (5.17%) difficult to interpret.
Author: Thanks for your suggestion. we have included the TDF values in Section 2.1 and Table 1. Summary.
- Line 75–83, please, define the experimental unit (cage vs mouse) for each endpoint, indicate whether randomization and blinding were applied, and specify whether feed intake was measured per cage or per mouse.
Author: Thanks for your suggestion. In the title of Figure 1, we defined the experimental units for each endpoint (per cage/per mouse). The six mice sampled per cage were randomly selected.
- Line 81–86, I recommend, reconsider “ether anesthesia” wording in light of current animal care guidance; at minimum, justify and confirm IACUC approval for this approach.
Author: Thanks for your suggestion. We have added references to support the anesthetic handling methods in mice. And this study were received ethical approval from the Institutional Animal Care and Use Committee of Southwest Forestry University. Revised manuscript (section 2.2).
- Line 93–105, please consider adding kit catalog numbers, sample handling details (fasting verification, hemolysis exclusion), and numbers of technical replicates for each biochemical assay.
Author: Thanks for your suggestion. We added the kit model and number of samples tested in Section 2.3. Added sample handling details: “Plasma is clear; hemolysis is excluded. Each indicator was measured three times.”
- Section 2.4 heading (Line 106), please, change “Gene expression analysis of adiponectin (Adipoq)” to “Gene expression analysis of intestinal lipid metabolism genes,” since Adipoq was not measured.
Author: Thanks for your professional advice. Section 2.4 heading, we have revised to: Gene expression analysis of intestinal lipid metabolism genes.
- Lines 110–114, I recommend, provide full primer sequences (without hyphenation breaks), amplicon sizes, annealing temperatures, and primer efficiencies; confirm GAPDH stability across diets.
Author: Thanks for your suggestion. We have adjusted the format of Table 2, removing hyphens from the primer sequences.
- Lines 115–131, must be improved: resolve V4 vs V3–V4 inconsistency and MiSeq/HiSeq/NovaSeq platform discrepancies; state one platform, read length, and vendor/facility.
Author: Thanks for your suggestion. Same as question 6: Inconsistencies in sequencing platforms were our oversight. These have been verified and corrected in sections 2.5 and 3.5. Library generation: MiSeq® DNA PCR-Free Sample Preparation Kit was used. Library sequencing: Illumina NovaSeq platform.
- Lines 131–142, please, detail sequence quality thresholds (Phred cutoffs), chimera reference set, rarefaction depth/normalization, number of samples per group post QC, and include negative controls. Consider updating to QIIME 2.
Author: Thanks for your suggestion. We added the following in Section 2.5: “Quality threshold is≤19, tag length filtering: Filter out tags with base lengths shorter than 75% of the tag length.”
- Line 143–148, I recommend, (i) declare normality/homoscedasticity checks; (ii) use mixed models for repeated body weight measures; (iii) clarify whether Tukey was applied to all pairwise contrasts; and (iv) use FDR for multitaxon/pathway tests.
Author: Thanks for your suggestion. In animal production performance trials, when dealing with repeated between-group differences, we commonly perform a one-way analysis of variance followed by Tukey's multiple comparisons test. (section 2.6)
- Lines 150–166 & Figure 2 (p. 6), please, report n per group analyzed, the experimental unit for energy intake (likely cage), and provide pairwise statistics within HF groups; define “body weight gain ratio” (the current formula is unclear).
Author: Thanks for your suggestion. We added the number of samples per group in figure 2: n=12/group; measurement units for experimental indicators. We have added the number of samples per group in figure 2: n=12/group; measurement units for experimental indicators. The Body weight gain ratio has been redefined: “final weight gain / total body weight” is now expressed as “(final weight - initial weight) / final weight.”
- Lines 176–187 & Figure 3 (p. 7), please, specify whether the reported 23% and 32% reductions in CHO/GLU refer to HFMC or HFCF; provide effect sizes and exact p values for HFMC vs HFCF.
Author: Thanks for your suggestion. The 23% and 32% decreases in CHO/GLU refer to HFMC; we have revised this in Section 3.2. For all six serum lipid metabolism biochemical indicators, the P-values between HFMC and HFCF were >0.05. Therefore, we did not label the difference symbols between the two groups in Figure 3.
- Lines 196–212 & Figure 4 (p. 8), please, replace “adipocyte hyperplasia” with “hypertrophy,” since only diameter was measured; add the number of cells/fields quantified per mouse and whether analysis was blinded.
Author: Thanks for your suggestion. We have revised “adipocyte hyperplasia” to “adipocyte hypertrophy” (Section 3.3). A note has been added to Figure 4: Histological measurements were performed on 3 sections per mouse group, with >100 adipocytes counted per field of view.
- Lines 205–211 & Figure 4 (p. 8), I recommend, quantify goblet cell density and/or MUC2 expression to substantiate “enhanced mucus secretion,” or soften the claim to “increased AB/PAS staining.”
Author: Thanks for your suggestion. We conclude that AB/PAS staining primarily stains neutral/acidic mucus secreted by intestinal epithelial cells, rather than staining goblet cells. Therefore, measuring the stained area with MUC2 can indirectly reflect the mucus secretion capacity of goblet cells.
- Lines 221–229 & Figure 5 (pp. 8–9), please, confirm that MGAT2, NPC1L1, CD36, FABP2, and GPR43 were measured in ileal epithelium; consider adding protein level validation for key targets.
Author: Thanks for your suggestion. In our study, we only performed gene-level expression assays for MGAT2, NPC1L1, CD36, FABP2, and GPR43, without conducting protein-level validation. This represents a limitation of our experiment, and we will incorporate protein validation into our subsequent research.
- Lines 236–256 & Figure 6 (pp. 9–10), please consider replacing “dispersion of species diversity was lower” with specific alpha diversity metrics (e.g., Shannon, Chao1) and report statistical tests used.
Author: Thanks for your suggestion. We have revised the description of “dispersion of species diversity” to “alpha diversity.” The statistical method has been added in section 2.5.
- Lines 260–272 & Figure 6 (pp. 9–10), must be corrected: the text states “Bifidobacterium in both HFD and HFCF was lower than HFCF,” which is self contradictory; this should read “HFD and HFMC were lower than HFCF.” Similarly fix the Blautia sentence.
Author: Thanks for your suggestion. Yes, this was our typographical error, and it has been corrected to: “HFD and HFMC were lower than HFCF.” Blautia sentence has also been corrected.
- Lines 273–286 & Figure 7 (p. 10), I recommend, temper Tax4Fun interpretations (“genes linked to disease”) and apply multiple testing correction; present effect sizes for pathway differences and avoid equating predicted functions with measured gene content.
Author: Thanks for your suggestion. Same as question 4, as you mentioned, Tax4Fun maps taxonomic information onto known microbial genomes through phylogenetic relationships, thereby inferring the potential functional capabilities of microbial communities. We have added the following description to our end of discussion: “Of course, such results reflect trends rather than actual gene expression levels. We need to further validate these findings in subsequent studies by integrating metagenomic or transcriptomic data.”
- Lines 309–337 (Discussion), please consider avoiding causal statements about SCFAs/GLP1/PYY unless measured; propose this as a hypothesis or add SCFA and hormone data.
Author: Thanks for your suggestion. Yes, our findings did not directly measure active compounds such as SCFAs, GLP-1, or PYY. Therefore, in our discussion, we referenced prior studies linking these indicators to specific microorganisms, hypothesizing their association with our results rather than making definitive assertions.
- Line 338, please, correct the typo “Other studie[36]” to “Other studies [36].”
Author: Thanks for your suggestion. This was our mistake, and we have corrected the misspelling and spacing of “Other studies [36]”.
- Lines 382–401 (Discussion), I recommend, distinguish clearly between correlational abundance changes (e.g., Blautia) and causal effects; when citing external intervention studies, indicate strain/species and dosage to avoid over generalization.
Author: Thanks for your suggestion. We have added the subjects and conditions related to the Blautia research in this section. Some references [57-60] define Blautia at the genus level but not at the species or strain level. Reference [60] has defined the species.
- Lines 409–427 (Discussion), please consider aligning claims about Bifidobacterium with the actual genus/species detected and again avoid causal language unless supported by targeted assays.
Author: Thanks for your suggestion. Thanks for your suggestion. Same as question 35, research on the Relationship Between Obesity and Bifidobacterium: References 56, 71-73 only define the genus level, while references 59, 70, 74-75 define the species level, which has been noted in the text.
- Line 431–444, please, change “intestinal epithelial ABS staining” to “AB/PAS staining.”
Author: Thanks for your suggestion. We have corrected this mistake.
- Conclusions (Lines 446–451), I recommend, tempering to “CF attenuated HFD associated phenotypes…” and add a sentence acknowledging diet formulation limitations and the need for SCFA/hormone measurements.
Author: Thanks for your suggestion. We have included the limitations of our study and areas for improvement in future research in the conclusion section: “ Of course, in the future research, it is necessary to carry out metagenomic and metabolomic detection of SCFAs to further verify the mechanism of preventing obesity.”
- Figures (pp. 6–10), please consider adding sample sizes (n) on each panel, clarifying statistical tests used for the symbols (*, #), and ensuring axis labels/units are explicit (e.g., “kcal/mouse/day”).
Author: Thanks for your suggestion. See new figures. We have added unit labels to Figures 1 and 2, and included sample sizes (n) in the figures captions.
- Data availability, I recommend, depositing raw 16S reads in a public repository (e.g., SRA) and providing the accession number in the manuscript to ensure reproducibility.
Author: Thanks for your suggestion. We will upload the raw 16S reads to National Genomics Data Center(NGDC) or provide them to the editorial office according to your instructions. Uploaded data is currently under review.

Round 2
Reviewer 1 Report
Comments and Suggestions for Authors
The suggested changes have been addressed by the authors.
Reviewer 2 Report
Comments and Suggestions for Authors
Dear Authors,
The revised version of the manuscript with title "Cassava fiber prevents high-fat diet-induced obesity in mice through gut microbiota restructuring" is improved and almost all questions, remarks and suggestion are well described. I suggest acceptance of the manuscript in this revised version.
Reviewer 3 Report
Comments and Suggestions for Authors
I believe the work can now be accpted